# Exploratory Study on the Impact on Emotional Health Derived from Participation in an Inclusive Choir

**DOI:** 10.3390/healthcare12131355

**Published:** 2024-07-07

**Authors:** Borja Juan-Morera, Icíar Nadal-García, Belén López-Casanova, Lucía Estella-Escobar

**Affiliations:** 1Department of Musical, Plastic and Corporal Expression, University of Zaragoza, 50009 Zaragoza, Spain; iciarnad@unizar.es; 2Education Department, Aragón Government, 50004 Zaragoza, Spain

**Keywords:** mental health, emotions, choral singing, inclusivity, social inclusion, emotional impact, personal development, well-being, music, choir

## Abstract

Participation in group activities such as choirs has been shown to have positive effects on emotional health and overall well-being. Inclusive choirs, which integrate individuals of various abilities and diverse backgrounds, provide a unique space for social interaction, emotional expression, and inclusion. This study aims to explore the impact of participation in an inclusive choir on the emotional health of its members, identifying both positive and negative emotional impacts as well as personal experiences derived from their participation. This longitudinal exploratory study combines participant observation, field notes, focus groups, and questionnaires to gain a deep understanding of the participant’s emotional experiences through their narratives. The study was conducted in an inclusive choir located in a medium-sized city in Spain, which brings together people of various ages, genders, abilities, and cultural backgrounds. The results indicated that most participants experienced significant improvements in their emotional well-being, including increased self-esteem, a greater sense of belonging, and reduced symptoms of anxiety and depression. Participants also reported that the choir provided a safe space for emotional expression and the building of meaningful relationships. Participation in an inclusive choir can have a considerable positive impact on the emotional health of its members.

## 1. Introduction

Participation in group activities such as choirs has been shown to have positive effects on emotional health and overall well-being [1]. Inclusive choirs, which integrate people of diverse abilities and different backgrounds, offer a unique space for social interaction, emotional expression, and inclusion. However, there is a lack of specific research on the emotional benefits of these inclusive choirs.

Choral music has indisputably become the most prominent genre of collective music-making globally [2]. According to [3], the practice of music stimulates the brain connections that activate emotions and collaborates in the process of psychic and affective development that facilitates the right balance to achieve the improvement of well-being. Engagement with music improves one’s perception of oneself, but only if it provides positive learning experiences that are rewarding [4].

Positive feelings such as joy, love, sense of accomplishment, belonging, appreciation, and respect are a necessary and indispensable part of artistic work [5]. In addition, according to different authors, the relationships between choral activity and quality of life have an impact on stress management, mood alterations, and the expression of emotions [6,7].

For [8], emotional perception does not have to be good or bad; however, in the educational environment it is possible to evaluate it in positive or negative terms. The causes and consequences of this emotional perception affect other psychosocial processes, such as well-being, motivation, inclusion, and social justice, among others.

However, it is worth bearing in mind that it is not an easy task to establish a reliable correlation between active musical practice and the acquisition of emotional skills, as measuring the occurrence of emotions is challenging [9].

This study focuses on the Cantatutti Inclusive Choir (CIC), founded in October 2017, at the Faculty of Education of the University of Zaragoza. The project was created with the aim of promoting inclusion and musical accessibility in the socio-cultural and educational context of the city [10]. This choir, since its conception, has been a reflection of cultural, generational, functional, and socioeconomic diversity, among others, welcoming members of both the university community and society in general. It is a musicosocial program that combines the characteristics and inspirations of community music [11], social action through music [12], musicking (music making) [13], and music education activism [14] in the Spanish city of Zaragoza. One of its main characteristics is the polyphonic signing of music in Spanish sign language [15].

The context of the research includes the choir participants and their immediate environment. These people form a heterogeneous and diverse group in terms of characteristics and previous experiences; they are of varied ages and backgrounds and may or may not have a background in choral singing or musical training. Nevertheless, they all have something in common: a shared passion for music, alongside a deep-seated sensitivity to the imperative of social inclusion.

The city of Zaragoza, where the choir resides, has an important musical and choral tradition, which implies a rich history and sensitivity towards the cultural heritage related to music and singing. In addition, the choir focuses on social inclusion through music, which places it in a context of a constant search for equal opportunities and musical educational activism.

## 2. Materials and Methods

This is an exploratory longitudinal study that combines participant observation, the field notebook, focus groups, and a questionnaire to gain an in-depth understanding of participants’ emotional experiences through their narratives [16,17].

Data collection was carried out using a variety of qualitative techniques and instruments, consistent with the principles of the study. These include:Participant observation: direct observation and participation in CIC activities allow a deeper understanding of the dynamics of the choir, the interactions between its members, and the context in which it operates [18,19].Field Notebook: the chronological records of all the events and observations made since the beginning of the activity are collected in an instrument that offers a longitudinal perspective of the CIC and that broadly covers the different environments, nuances, and specificities in which the project takes place [20].Focus groups: Focus groups with choir members offer detailed and nuanced perspectives on individual and collective experiences [21,22].Questionnaires: The implementation of questionnaires helps in the compendium of qualitative information that enriches and triangulates the findings of the study [20].

The analysis was carried out through an iterative and reflective process [23], where data were encoded and interpreted for meaningful patterns, themes, and relationships using specific software [24]. To organize and facilitate the handling of large volumes of data, a qualitative analysis software was used (Nvivo 14.23.2). This approach offers a deep and nuanced understanding of CIC and respects the complexity of its realities and experiences. 

Reflexivity is central to the chosen methodology. Researchers maintain a critical and analytical attitude, recognizing how one’s own experiences, preconceptions, and individual position can influence the research process and the interpretation of data. In addition, rigorous ethical principles are established, ensuring confidentiality, informed consent, and respect for all participants [25].

In this sense, the analysis has been validated through continuous discussions with supervisors, university peers, and at international music education conferences. An attempt has also been made to establish a dialogic intersubjectivity in relation to the participants. Validity in qualitative research involves considering the degree to which the informants’ views, thoughts, feelings, intentions, and experiences are accurately understood by the researchers and portrayed in the results report.

A variety of strategies have been applied to address the validity and reliability of the findings. To ensure validity, low-inference descriptors have been used, discussing the results with other researchers as well as with the participants of the choir; participating investigators used mechanical means for the recording, storage, and retrieval of the collected data [26]. 

A prolonged stay in the field strengthens research reliability [22]. A persistent observation of the dynamics and practices of the choir has been carried out, and triangulation of methods and sources has been applied. In addition, a check has been carried out by the members of the choir to validate the intentionality, correct errors, and obtain additional information that enriches the understanding of the phenomenon studied. In the case of focus groups, this usually means letting informants read their transcribed interventions to confirm that it is correct or involving participants in the discussion of the analysis itself [27]. 

It is relevant to refer to the fact that, although some members of the choir are not familiar with analytical perspectives, an effort has been made to ensure that the experiences and statements discussed in this research project are faithful to the way in which the participants understand them. These practices have allowed the authors to ensure the coherence and reliability of the events and situations observed in the context of the choir.

The aim of this study is to explore the impact on the emotional health of participants in an inclusive choir, identifying the positive and negative emotional impacts as well as the personal experiences derived from their participation.

In order to obtain a deep understanding of the phenomenon studied in this research, a phased scheme is designed to facilitate the collection and analysis of the following data:Participant observation phase: In this phase, an immersion in the context of the CIC has been carried out over the last six years (2017–2023). During this period, all the activities carried out have been recorded in the field notebook, maintaining an active and participative observation. To this notebook have been added the most significant qualitative contributions, collected in 58 forms filled in by the choristers since the beginning of the musical activity in 2017, and the most important interventions of the official choir group on WhatsApp (2.24.12.78) since its creation on 5 December 2017, as well as a selection of personal messages addressed to the project coordinators by these same participants and their environment.Design and validation phase of data collection instruments (questionnaires, focus groups): These instruments have been individually designed or adapted and have been subjected to a validation process through expert analysis or the use of specific software and pilot application in a small group of participants. The objective of this phase has been to ensure the reliability and validity of the data collection instruments in order to guarantee the quality of the information obtained.Phase of carrying out questionnaires to the choir and its environment: Two structured questionnaires have been administered, the first aimed at the group of participants and the second at their environment. Both questionnaires were managed through an online platform that made it possible to obtain useful information on socio-demographic profiles, perceptions, and opinions about the project. This phase has made it possible to broaden the scope of the research by obtaining qualitative information.Focus group phase: Two focus groups have been carried out to obtain information about the dynamics and interaction of the group, as well as to learn about their motivations, experiences, perceptions, and opinions about the project.Data analysis phase: Following the previous phases, a study of the data obtained has been carried out using an open and axial coding technique that identifies emerging patterns and categories through the Nvivo analysis software (14.23.2).

The phased methodological design has facilitated data collection from diverse sources and across various time points, enabling a comprehensive analysis of the study phenomenon. This approach ensures the quality and validity of the obtained results.

The study was conducted in an inclusive choir in a medium-sized city in Spain, bringing together people of diverse ages, genders, abilities, and cultural backgrounds.

The research includes 394 inclusive choir members who have participated in the project from its inception to the present and who represent a balanced representation of different demographics and levels of participation in the choir. At the same time, the testimonies of 219 informants from the environment with a degree of direct kinship were collected, mostly couples, fathers/mothers, children, siblings, grandparents, followed by friends and other relatives such as aunts and uncles and cousins.

As in any research, it is important to identify, narrow down, and describe the population to be studied and then select a sample that adequately represents that population in order to obtain relevant and accurate information [20]. For the authors, in research, the population or universe of study refers to the set of elements, whether people, families, groups, organizations, objects, recordings, performances, or any other entity that is intended to be analyzed in order to obtain relevant information. 

In the case of the population analyzed in this research, it is all those current and previous participants who have passed through the CIC and its closest environment. In other words, the overall sample of this study is composed of the sum of the individual samples of each instrument used for observation and data collection. This includes both active participants and those who have ceased to be part of the choir at some point, as well as their family, friends, and acquaintances who have had direct contact as spectators or receivers and can provide relevant information. It has been considered that the sample is representative and as diverse as possible in terms of age, gender, origin, musical experience, and socioeconomic origin, with the aim of obtaining a vision, as complete and diverse as possible, of the beliefs of those involved in the activity.

The questionnaires, designed ad hoc for the choir, were sent to be completed by all participants registered in the project’s database (current and former), which amounted at the time of submission, in January 2023, to 344 people, with 155 active participants and 189 former participants. Of these, 185 responses were received (54% of the population) with 113 responses from active participants (73%) and 72 responses from former participants (38%). At the same time, it was the members themselves who were in charge of disseminating the survey “for the environment” among their relatives, from which 219 responses were collected. The questionnaire addressed key dimensions such as demographic information, perceived benefits, inclusion, social transformation, and areas for improvement. Specific questions were designed to elicit detailed responses, such as “In what ways do you think participation in the choir improves the lives of the participants, based on your own experience?” and “Does participating in the choir provide you with emotional growth?”.

To carry out the focus groups, the sample has been selected through an intentional sampling process, which involves the identification and selection of participants who meet the criteria of representativeness and diversity, with the aim of obtaining a complete and diverse view of their beliefs about the activity. The sample was composed of 16 individuals divided into two focus groups, each with 8 participants.

In alignment with the journal’s ethical policies, the research was conducted with the utmost responsibility and respect for the participants, strictly adhering to the highest ethical standards. All subjects provided informed consent prior to their participation. Throughout the study, participants’ confidentiality and anonymity were preserved, following ethical principles of respect, beneficence, non-maleficence, return of results, and justice. The study meticulously followed protocols recommended by academic literature and the Declaration of Helsinki concerning research involving human subjects [25].

Moreover, it is important to highlight that the CIC, where the research was conducted, operates under the Music and Inclusion for Social Change Chair. This chair was established through a partnership between the University of Zaragoza and the Institute of Social Services of the Government of Aragon. This collaboration underscores a commitment to research integrity and ethics, particularly in projects promoting social inclusion through music and focusing on the wellbeing of vulnerable and disadvantaged individuals.

Since the University of Zaragoza lacks a specific ethics committee, special attention was given to ensuring adherence to the ethical guidelines set by the Music and Inclusion for Social Change Chair, which prioritizes ethical research among its core objectives.

## 3. Results

The results obtained throughout the research are detailed according to the different instruments used during the exploratory process. 

For a rigorous and detailed analysis, the data collected has been processed using Nvivo software, a tool specialized in the management of qualitative information. Through this software, it has been possible to organize the information into several thematic nodes, which has facilitated a more structured and comprehensive interpretation of the data. Finally, the results have been triangulated from each other to ensure their validity and reliability, as well as to draw the most relevant conclusions of the study. 

Regarding the different references coded and shown hereafter for each thematic node, depending on the research instrument, a common selection criterion has been followed based on its significance. Thus, some of the most representative recorded references have been exposed, which illustrate the different sub-nodes contained by the nodes or higher categories. Redundant, decontextualized references or lack of meaningful content have not been included in the writing of this article in order to facilitate the reading and understanding of the data. The classification and details of these nodes are presented in the table below (Table 1). However, the study focuses on nodes 1.2, Positive Emotional Impact, and 1.3, Negative Emotional Impact, which correspond to macronode 1, Personal Development and Well-being.

Since the informants in this study comprise more than 600 individuals, no anonymization has been carried out by assigning fictitious numbers or names. When a dialogue or conversation excerpt appears, it is transcribed as P1, P2, etc. In the case of proper nouns, which appear in a sentence that loses its meaning without its mention, the name is replaced by X, Y, Z, etc. Some interventions have been modified to ensure readability, as well as to ensure anonymity in cases where the written or oral expression of the informant can identify the subject. For the same reason, interventions that may evidence or stigmatize some type of (dis)ability or the cultural level of the participants have been corrected. Sometimes, only if the understanding of the context conditions the understanding of a contribution, some characteristics of the subjects are outlined, but never with the intention of labeling, but with the aim of contributing to a deep understanding of the information.

### 3.1. Field Notebook

The field notebook collects 758 entries chronologically from 1 September 2017 to 15 September 2023. This instrument reflects the participants’ observations of the authors throughout the process, the significant excerpts from the WhatsApp group, the transcripts of the intervention of different participants in different public events, and the qualitative responses of informative value for the research, contained in the 85 forms that have been filled in by the individuals involved in the CIC to date.

Several data related to CIC activities are collected in the field notebook; 44 performances were recorded, reflecting the group’s active participation in different events. 

The booklet also includes information on 17 extra-musical experiences, such as exchanges, collaborations, and coexistence events, both internal among the members of the project and external with other collectives. These activities highlight the choir’s interaction with a wider community. 

The field notebook, as well as the implementation of 25 training workshops, was aimed at the development of specific skills and knowledge of the choir members. 

The nodes referring to the positive and negative emotional impact encoded through the Nvivo software offer a distribution in terms of the source of information, as shown in Figure 1.

#### 3.1.1. Positive Emotional Impact

There are 247 references encoded in the field notebook, with a coverage of 16.06%. They refer to testimonies that express happiness, that describe enriching experiences inside and outside of rehearsals, and that recognize individual or collective contributions and impact through performances. Some of the most significant notes or testimonies of the participants collected in the field notebook, are as follows:
Reference 1: I would never have imagined that I would be able to sing in public; I thought I was incapable; it’s amazing that we have done this in 2 months; it has been the best decision of this course, etc.Reference 2: I went to the emotions session the other day and I liked it a lot, and I have been encouraged to join.Reference 3: The best decision I’ve made this course!Reference 4: From the first rehearsal I participated in, I knew I would still be part of the choir. It’s totally fantastic and I encourage anyone who wants to have a unique experience to join the choir.Reference 5: Not even in my best hopes did I imagine that I would find a choir where I could sing and above all a group of humble, funny people, always gentle and willing to help. Good people!!Reference 6: I want to make it clear that if I hadn’t met the choir, I wouldn’t have taken anything positive away from 2018. It’s been a very bad year for me, in many ways, and the choir has helped me relax during that period when we were singing together.Reference 7:P1: How beautiful it is to convey emotions with your voice or gestures. A kiss to all.P3: Totally agree!! Yesterday with the final applause some tears escaped. P1: It’s an amazing feeling to sing together at a concert, despite the tension of not controlling much. It’s worth it, no doubt.Reference 8: P1: It’s amazing how you can feel the energies when you sing all together, it creates something difficult to explain.P2: Congratulations guys!!!! I have enjoyed a lot, we have sounded great, the gestures are great, and the soloists, presenters, and pianists and conductors and everyone in general!! It is a pleasure to do these things with you!! 
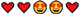
P3: I say the same thing!! What a rush of positive energy and how lucky to be a part of this. I feel happy and grateful, and I also really believe that we are sending a very valuable message to society. 
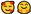
Reference 9: (Testimony of the artist and singer Blas Cantó during his visit to a choir rehearsal) I thank you, because when I came to your rehearsal I didn’t know what I was going to find. I think the most important thing is not just to sing well, it’s to transmit. And you have touched my heart. You’ve reminded me why I love music. Really, thank you very much, keep doing what you do, because it’s special.Reference 10: How happy X. Thank you 
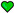
 so much. These are the happiest moments of the week. What a success, thank goodness we got 
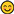
 it.Reference 11: It’s been a long time since I’ve been to a show, a performance by someone who made me feel that way. I wish all my friends, all my family, everyone had been there to see it. Because no matter how much I tell it, I can’t convey what I felt and how I felt it. We sang the first songs crying, holding hands, my husband and I, because you know what any of us parents who have a child with a problem go through. And of course this is much more than a choir.Reference 12: Going to Coro is always for me a time to leave behind the problems of the day and enjoy yourself.Reference 13: A place where you feel at ease, you feel like you can be you, you don’t get judged and most importantly, we have a good time, we make music and we enjoy ourselves.Reference 14: It has given me a lot of strength and positive energy. Emotionally, it has been very good for me.Reference 15: The emotional part is the one that I have noticed the most, because for me it has been a complicated year, and although sometimes it was difficult for me to go to rehearsals, I always leave there with a smile…Reference 16: It was amazing… both for me and for the people who went to see it. Everyone left delighted, wanting to come to the next one. For me it has been an unforgettable experience. I still feel the “high” when I think about it. Impressive!!!Reference 17: When I remember the concert, I do so with admiration and satisfaction. The involvement that we all had to make it go perfectly, but above all I am left with the face of emotion that we all had after finishing the concert.Reference 83: The choir transmits happiness to me and I feel very comfortable and very good in every rehearsal or formal and informal act I do with the Choir. Let’s hope this never ends.

#### 3.1.2. Negative Emotional Impact

In the field notebook there are 22 coded references, with 1.22% coverage. These notes or testimonies refer to situations of stress, overwhelm, or self-demand with the activity; sadness at leaving the choir; interactions between participants; negative self-perception; or feeling of non-inclusion. The most representative are as follows:
Reference 18: I’m very sorry that I was almost able to attend the rehearsals during this last month, but the truth is that between these 4 h of class a day and at the same time the end of the course in Education… I feel crushed by work and I’m not in a good moment.Reference 19: Due to stress problems that I can’t control, I have to leave the choir. I don’t know if I’ll come back one day, but now I can’t go on very much thank you all, see you.Reference 20: Hello, I’m leaving the choir; I can’t cope with this maelstrom of emails, songs, and projects. I’m too old; if after Lithuania the rehearsals are quieter, if possible, I’ll sign up again.Reference 21: I’m 57 years old; sometimes when people say “this is for young people, not for you”, I feel a little sad.Reference 22: My biggest fear is not being able to reach the level that the choir has or not being able to give everything that is expected of me.

### 3.2. Choir Questionnaire

In the questionnaire applied to the members of the CIC, a series of responses to qualitative questions have been coded, designed to explore various facets of their experience in the choir. A total of 1811 nodes have been coded from the 185 responses received to the instrument. 

The questions cover topics such as the perceived improvements in the lives of the members as a result of their participation in the project and any negative effects experienced during their involvement. It has been investigated how participation in the choir may have modified beliefs or preconceived ideas on the part of the members, and the values promoted through the initiative have been identified. Finally, questions were asked about perceived difficulties in the development of CIC activities, and respondents had the opportunity to suggest proposals for improvement and were invited to share any additional comments or reflections inspired by their experience at the choir or by the questionnaire itself. The codes referred to the emotional impact are shown in Table 2.

#### 3.2.1. Positive Emotional Impact

There were 102 references coded in the questionnaire to the choir, with a coverage of 5.47%. They mention testimonies that express happiness and describe enriching experiences inside and outside the rehearsals. Some of the most significant records of the participants, collected in the questionnaire to the choir, are the following:
Reference 23: I never thought that singing in a choir would be so dynamic and fun.Reference 24: Not only has it taught me about the artistic field, it has also given me knowledge on an emotional level.Reference 25: Singing excites me, gives me joy.Reference 26: It is my personal therapy of life.Reference 27: Years ago, when I was younger and my OCD wasn’t on the rise, I was able to do a lot of activities, including singing at school with classmates, and again with Cantatutti to be able to take up this activity again with people who now understand my limits is great.Reference 28: I feel more alive!!Reference 29: My mind is calmer on a daily basis as I find myself humming the songs we rehearse instead of dwelling on my worries.Reference 30: Emotional management in periods of high stress and loneliness.Reference 31: In my personal well-being, by doing something I like, feeling fulfilled, well surrounded, disconnecting…Reference 32: In the emotional realm. It helps me stop, disconnect from the stress of everyday life and reconnect with myself.Reference 33: Increase my self-esteem and self-confidence, after the pandemic and the opposition period.Reference 34: I’m more excited about doing things.Reference 35: It has changed my life completely; the choir makes me happy and makes me have that motivation and joy that I had lost.

#### 3.2.2. Negative Emotional Impact

In the questionnaire to the choir there are 23 coded references, with 1.24% coverage. These notes or testimonies allude to situations of stress, overwhelm, or self-demand with the activity, interactions between participants or negative self-perception. Some of the most representative are set out below:
Reference 36: As the level increased, I demanded a lot from myself (myself) and that made me uncomfortable.Reference 37: I’m embarrassed to answer that I’m not important to the project; that’s the only thing I answered 0, because sometimes I think that if I don’t go it doesn’t matter.Reference 38: I think it has increased my stress level from having more activities in my schedule and decreased my attendance at college classes quite a bit (although it’s still not fair to blame this on the choir).Reference 39: Not all people have the same emotional and temporal availability to dedicate to social issues or issues that matter to us, as was my case (family responsibilities, care burdens, work stress, chronic anxiety…) Although for a while my social anxiety improved, the last few months that I participated it increased tremendously.Reference 40: At some point I am overwhelmed by trying to achieve everything.Reference 41: Some anxiety before the performances.Reference 42: Not doing it right weighs me down.Reference 43: Some stress before concerts and in rehearsals of pieces that I find more difficult, but nothing major. What gives me a hard time is learning new letters in languages I don’t understand without seeing them written.Reference 44: Frustration at not knowing the songs well, when I don’t study them.

### 3.3. Environment Questionnaire

In the questionnaire addressed to the environment of the members of the CIC, various answers to qualitative questions have been collected, focusing on the perception of this group about the experience of the participants. A total of 821 nodes from the 219 responses received to the instrument have been coded.

The questions inquire about whether participation in the CIC improves, in their opinion, any aspect of the lives of its members, or on the contrary produces some negative effect. Their proposals for improvement, the possibility of including other issues that they consider decisive for the questionnaire, or the initiation of any type of debate or reflection as a result of the questionnaire are taken into account. The codes referred to the emotional impact are shown in Table 3.

#### 3.3.1. Positive Emotional Impact

There were 26 references coded in the questionnaire to the environment, with a coverage of 1.32%. These refer to enriching experiences of the participants. Some of the most significant responses, collected in the questionnaire to the environment, are as follows:
Reference 45: Yes, the choir has been a great support emotionally.Reference 46: I am more than 12,000 km away. It makes me sensitive to see the work they do… They are unique in the world.Reference 47: She is happy and very grateful to be part of the choir.Reference 48: Helping people who have a hard time with social relationships improves their self-esteem.Reference 49: Makes participants and audience happier.Reference 50: It has helped some friends to expand their circle of friends. In addition, they go to rehearsals with enthusiasm, which can improve their mood day by day.Reference 51: Is happy participating.

#### 3.3.2. Negative Emotional Impact

In the questionnaire to the environment, there are 4 coded references, with 0.20% coverage. These notes or testimonies allude to situations of stress, overwhelm, or self-demand with the activity. The most representative are as follows:
Reference 52: Stress at specific times.Reference 53: She gets a little frustrated with sign language, but she tries really hard and wants to do it right.

### 3.4. Focus Groups

In the two focus groups carried out with members of the CIC, a series of references derived from the dialogue and discussion of the participants have been compiled. It should be noted that the coding and the thematic nodes identified have been consistent in both groups. A total of 717 nodes from both sessions have been coded.

The questions addressed various aspects of the experience of the participants in the CIC. It explored how they describe their experience from the beginning to the present, the perceived changes in their lives since joining the initiative, and how they feel about the contribution of the CIC to the promotion of inclusion and diversity in society, as well as the impact that their participation has had on their perception and attitude towards diversity. In the same way, the strengths and weaknesses experienced in the project, the best experiences to date, and suggestions for improvement to enhance different aspects of the choir have been discussed. The codes referring to the emotional impact are shown in Table 4.

#### 3.4.1. Positive Emotional Impact

There were 33 and 34 references coded in the focus groups, with a coverage of 34.94% and 28.89%, respectively. They refer to testimonies that express happiness and that describe enriching experiences inside and outside the rehearsals. Some of the most significant notes or testimonies of the participants, collected in these, are shown below:
Reference 54: I don’t know how to say it… I get excited. In the choir you can be happy, when you have a disability… I do not know…Reference 55: On a personal level for me it has meant everything, that is, I have a new group of friends to go out with, no one would have told me in my life that I would learn sign language and that I could like it so much.Reference 56: The project itself is what it does, it involves you, you like it so much and it reaches you and fills you so much that you say wow, I want to participate in this, I want to help in this.Reference 57: I feel good, I think it’s fun, it brings me benefits, so, well, I decided to stay. And from the first day I said it: this is a nice environment so I stayed for that.Reference 58: A very good atmosphere was created and I’ve been here ever since.Reference 59: Well, the first thing for me is that emotionally it was already important, because I started in 2017 and at the beginning of the course I felt very lonely here, because I don’t know, I didn’t have anyone good, or almost no one, and when I started university I was very lonely.Reference 60: Then, as I said before, but I repeat, at first I felt very sad and lonely because I had arrived in Zaragoza a short time ago.

#### 3.4.2. Negative Emotional Impact

In the focus groups, 1 and 2 coded references appeared, with 2.31% and 0.22% coverage, respectively. The most striking are as follows:
Reference 61: I don’t feel rejected, but I don’t feel comfortable with festive recreational activities, because I feel out of place. Not because I’m shy or because I’m not; because what do I do at 2 a.m. in the morning? The average choir is younger than my children. [laughs]Reference 62: And it has happened at some point that many of us, at some point, have been burned.

## 4. Analysis and Triangulation

This section focuses on unraveling and understanding the patterns, themes, and emerging narratives identified through the different qualitative methods used: the field notebook, participant observation, questionnaires open to the choir and the environment, and focus groups.

Qualitative analysis focuses on interpreting the experiences, opinions, and perceptions of choir participants, as well as those indirectly involved in their environment. This approach aims to transcend numbers and statistics, to capture the essence of human experiences and the underlying social dynamics in the choral context. This section explores the meanings and relevance of emerging themes and how they connect to the goals of the study.

Regarding the frequency of nodes depending on the data collection instrument used, it is interesting to emphasize the presence of both nodes in most of the instruments with a balanced proportion, as shown in Figure 2.

In the coding matrix (Figure 3) resulting from the qualitative triangulation, the 36 thematic nodes provide a multifaceted view of participants’ experiences and perceptions. This matrix reveals how these nodes, though distinct, intertwine and influence each other, offering a holistic understanding of the choir’s impact on its members and the relationships between thematic categories. This chart allows us to understand how positive or negative emotional impacts interrelate with other fundamental aspects of the project reported by its participants and environment.

The remarkable coincidence between **personal development** and **positive emotional impact**, manifested in 209 points of intersection, suggests a significant correlation between individual choir members’ growth and their positive emotional experiences. This link implies that choir activities not only promote personal development but are also a source of emotional well-being, highlighting the importance of the choral environment in promoting a sense of satisfaction and fulfillment. 

Additionally, the **positive emotional impact** is strongly linked to the **sense of belonging and community**, with 204 coincidences. This shows how positive emotions are highly rooted in the feeling of being part of a choral community, suggesting that emotional well-being is intrinsically linked to the sense of connection and belonging to the group. Likewise, the relationship between **positive emotional impact** and **motivation**, indicated by 182 matches, emphasizes that positive emotional experiences are an important driver for participants’ motivation. This suggests that enjoyment and emotional satisfaction may be key factors driving participation and engagement in the choir.

However, the relationship between **negative emotional impact** and **challenges** within the project, although less frequent, with 18 coincidences, suggests that negative emotional experiences may be linked to certain perceived challenges. This could prove that problematic or difficult aspects of the choir can have an adverse emotional impact on some people involved in the initiative. Similarly, **negative emotional impact** and **high expectations**, with only 12 matches, suggest that, although expectations within the choir may be high, they are generally not a significant source of adverse emotions for members.

The connection between **inclusivity** and **positive emotional impact**, found in 171 matches, emphasizes the role of inclusivity in generating positive emotional experiences, which warns that inclusive initiatives, in addition to promoting equity and acceptance, are also an important source of emotional satisfaction and well-being for the people who are part of this initiative. In addition, the relationship between **accessibility** and **positive emotional impact**, with 99 matches, suggests that increased accessibility is directly related to positive emotional experiences within the choir. This indicates that accessibility facilitates enriching and emotionally rewarding experiences for participants.

The connection between the enjoyment and value of the musical experience and the positive emotional impact, with 148 coincidences, reveals that musical enjoyment is intrinsically related to the positive emotions experienced by the choir members. This means that the musical experience in the choir is not only musically rewarding but also emotionally enriching. In addition, the connection between **the expression of gratitude** and the **positive emotional impact**, with 51 matches, reveals that the positive emotional moments experienced in the choir foster feelings of gratitude. This highlights the influence of positive emotional experiences in creating an environment of gratitude and recognition within the choral project.

Finally, the analysis of the nodes referring to emotional impact is illustrated in Table 5, with different sub-nodes or categories that allude to the different types of narratives included in these nodes.

## 5. Discussion

The results converge in highlighting the significant impact of the CIC on the personal development and emotional well-being of its members. The analysis of the nodes, derived from sources such as the field notebook, questionnaires, and focus groups, reveals a remarkable influence of the choir on self-esteem, interpersonal skills, and the formation of a strong sense of community. This phenomenon resonates with existing literature that underscores the therapeutic and cohesive effects of choral music [6,7], evidencing the way in which the CIC functions as a space for emotional support and social development. While the study highlights the positive impact of participation in the choir on emotional well-being, it is crucial to clarify that it does not start from the premise that any positive emotion is intrinsically healthy [28]. Emotions are the basic neurological processes present in all people, and the performance of certain activities can generate positive emotions [29]. However, this does not imply that a positive emotion is inevitably healthy. The focus is on investigating how participation in a structured, social activity such as choral singing can influence the emotional health and well-being of participants.

The impact of the CIC on the personal and social development of its members is a significant finding. Their participation in the choir not only improves musical skills but also contributes to the development of social and emotional competencies. This choral experience acts as a catalyst to improve self-esteem, communication skills, and empathy, among others, aligning with research from [30], who highlight the role of musical ensembles in the development of identity and social skills.

The results show that participation in the CIC has a positive effect on the mental health and emotional well-being of the people who are part of the project. Choral singing, in this context, becomes a therapeutic tool that provides an escape route from daily stress and fosters a sense of accomplishment and satisfaction. These findings are in line with previous studies that have shown how participation in choral activities can improve psychological well-being and reduce symptoms of anxiety and depression [7]. Qualitative data from participants’ self-reported narratives indicate concrete improvements in their mental health thanks to the feelings and emotions generated by their participation in the choir. These findings align with existing literature [28,31], which suggests that group musical activities can improve emotional resilience and well-being when performed in a supportive environment.

In the field of education, choral programs should be designed to promote musical skills, but learning related to social and emotional competencies should not be neglected. Ref. [4] highlights the social and emotional benefits of music education; educators must integrate activities that promote empathy, cooperation, and conflict resolution, taking advantage of the diversity of the choral group as an educational resource. The importance of context and educational aspects in the therapeutic potential of musical activities is emphasized. The inclusive nature of the choir in the studio, which integrates individuals of diverse abilities and backgrounds, creates a unique social dynamic that amplifies the emotional benefits observed. The role of choir facilitators is crucial [32], as they provide the support and guidance needed to maximize these benefits. Therefore, it is highlighted that context and structure are critical to the positive outcomes associated with participation in the choir.

Socially, inclusive choirs can act as small representations of communities, reflecting, and at the same time influencing, broader attitudes towards diversity and inclusion. Attending to [33], choirs can be platforms for intercultural dialogue and mutual understanding. Choral projects have the potential to collaborate with mental health and wellbeing initiatives, providing a safe space for expression and emotional support.

For future research, the inclusion of control groups could be used to better establish causal relationships between participation in the choir and improvements in emotional health. In addition, the integration of quantitative methods together with qualitative approaches would provide a more complete understanding of the impacts. The use of standardized measures of emotional health could complement qualitative data, improving the robustness of the findings. A detailed exploration of any negative emotional impacts and their underlying causes would also help develop strategies to mitigate these effects, ensuring an overall positive experience for all participants.

The study provides promising insights into the emotional health benefits of participating in an inclusive choir. However, more research is needed to deepen the understanding of the mechanisms involved and validate these findings in broader contexts. By addressing these areas, one can better understand how inclusive musical activities can serve as effective interventions to improve emotional well-being.

## 6. Conclusions

The results found that most participants experienced significant improvements in their emotional well-being, including an increase in self-esteem, a greater sense of belonging, and a reduction in symptoms of anxiety and depression. Participants also reported that the choir provided a safe space for emotional expression and building meaningful relationships.

The positive emotional impact is a constant, where joy, satisfaction, and overcoming personal challenges through musical practice have been common experiences. Music, in this context, is not only experienced as a recreational activity but also as emotional therapy and a means of connecting with others.

In addition, in the “Positive Emotional Impact” node, coded references explore a range of positive emotions. Public performances emerge as moments of emotional climax, while activities outside of rehearsals reveal the richness of the choral experience beyond singing. Similarly, the node captures the essence of the personal interactions that have enriched the emotional well-being of the members, as well as the reactions of the public and the overall influence of the project.

On the other hand, the “Negative Emotional Impact” node focuses on aspects of participants’ experience that have had adverse emotional repercussions. Personal interactions that have turned out to be a source of stress or uneasiness are encoded here. The emotional complexity associated with leaving the choir is contemplated, as well as the feelings of exclusion or deterioration in self-perception, offering a balanced view of the emotional challenges faced by the members.

Therefore, participation in an inclusive choir can have a considerable positive impact on the emotional health of its members. The creation and support of more inclusive choirs is recommended as a strategy to promote emotional health and social cohesion in diverse communities.

## 7. Limitations and Critical Reflections

The contextual specificity of the CIC, focused on a particular location and composition, restricts the generalizability of the results to other inclusive choirs or musical contexts. This limitation, known in qualitative research as “transferability” [34], implies that the conclusions reached are not universally applicable, raising questions about their applicability in different cultural and structural contexts. It was not the objective of this research to generate transferable results but to explore in depth the complexity of the project, but in any case, it is imperative to point this out.

Despite their usefulness, questionnaires and interviews have inherent limitations. These instruments may introduce socially desirable response biases or subjective interpretations [35]. The inclusion of additional methods, such as prolonged observations or analysis of past conversations (WhatsApp), are intended to provide a more nuanced understanding and stronger triangulation.

On the other hand, the intensive and time-consuming nature of participatory observation, while providing a detailed understanding of the CIC, may also limit the breadth of perspectives collected. Immersion in a single group for an extended period carries the risk of leading to familiarity that potentially clouds objectivity and minimizes the observation of certain dynamics. A more frequent rotation between different groups or the inclusion of additional observers could have brought variability and avoided possible familiarity biases [36].

Given the researcher’s dual role as founders and choir coordinators, it is likely that unconscious biases exist in the interpretation of the data. Ref. [36] argue about the importance of reflexivity in study, where the researcher must be aware of their own perceptions and biases. Despite efforts to maintain objectivity, there is always a risk that personal and professional proximity to the project will influence the analysis. 

The interpretation of qualitative data is inherently subjective. While thematic analysis and coding of nodes provide a structure for understanding the data, the selection and representation of topics can be affected by the researcher’s perspectives and preconceptions. Triangulation with multiple sources and verification of findings with participants has helped mitigate these effects. 

The specific cultural and social environment in which the CIC operates may have shaped participants’ experiences and perceptions in ways that are not universally applicable. According to the authors’ statements, such as [37], culture plays a critical role in how people experience and make sense of their environments, suggesting that findings may be deeply rooted in the particular cultural context of the project.

Changes in choir composition and group dynamics over time represent a variable to be taken into account. Variability in the perception of stakeholders over time may influence the results, indicating that one-off inputs only capture a specific moment in time and may not capture the full evolution of members’ experiences in isolation [38]. In this case, since the research has been conducted over the course of six years, participants may have experienced changes in their personal lives as well as in their own opinions, a factor that implies that the transition in their perceptions is not fully captured in a single longitudinal study. Thus, further follow-up or studies could provide a more complete picture of these dynamic changes. In any case, the use of different data collection instruments, as well as the recording of the field notebook and the conversations shared by the participants in this period of time, provide an invaluable source that illustrates this complexity and enriches the results of this report.

The COVID-19 pandemic in 2020 brought unprecedented restrictions on choral activities, which could have significantly altered choir members’ experiences and group dynamics. Studies such as the one by [39] argue that music played a notorious role during the pandemic in Spain. But at the same time, any changes in CIC operations during this period, such as the transition to virtual activities, may have affected participants’ experiences and appreciations in ways that are not fully reflected in the study.

These limitations and critical thinking do not detract from the value of the study; rather, they offer a balanced and contextualized understanding of its results and provide a basis for future research. By acknowledging these barriers, the report aligns with the ethos of qualitative research, which values transparency, reflexivity, and ongoing commitment to improving and deepening knowledge.

## Figures and Tables

**Figure 1 healthcare-12-01355-f001:**
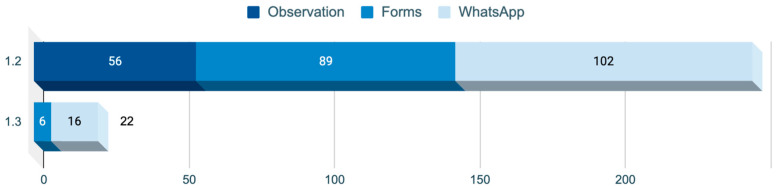
Distribution of data sources according to all 36 nodes. (Source: Author’s own elaboration).

**Figure 2 healthcare-12-01355-f002:**
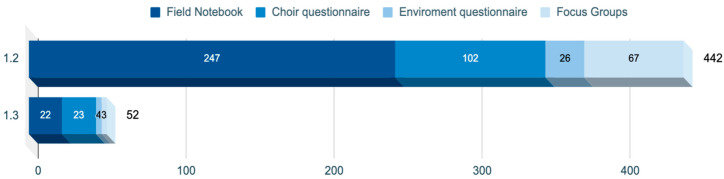
Distribution of nodes according to instruments. (Source: Author’s own elaboration).

**Figure 3 healthcare-12-01355-f003:**
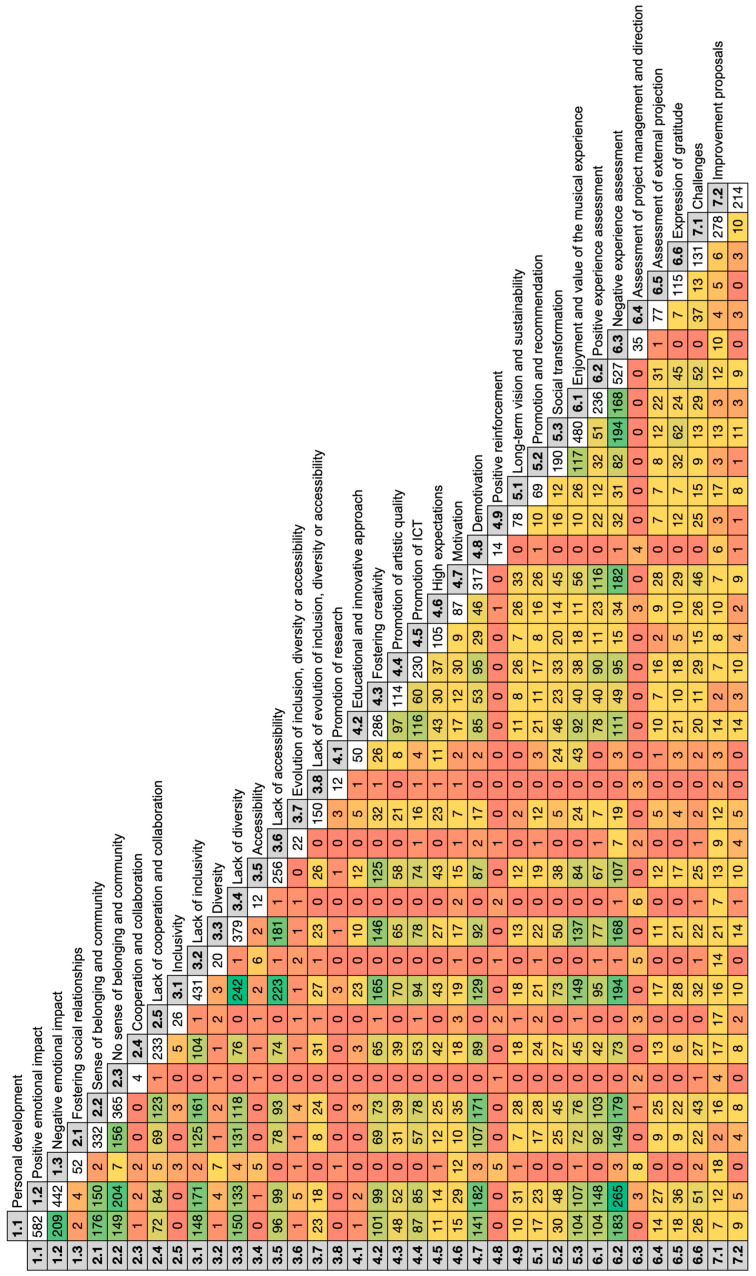
Coding matrix of the 36 nodes. (Source: Author’s own elaboration).

**Table 1 healthcare-12-01355-t001:** Hierarchical list of thematic nodes and their number of encoded records.

N	Nodes
6.951	Coding studio
1.076	1. Personal development and well-being
582	1.1. Personal development
442	1.2. Positive emotional impact
52	1.3. Negative emotional impact
960	2. Community & relationships
332	2.1. Fostering social relationships
365	2.2. Sense of belonging and community
4	2.3. No sense of belonging and community
233	2.4. Cooperation and collaboration
26	2.5. Lack of cooperation and collaboration
1.282	3. Inclusion, diversity and accessibility
431	3.1. Inclusivity
20	3.2. Lack of inclusivity
379	3.3. Diversity
12	3.4. Lack of diversity
256	3.5. Accessibility
22	3.6. Lack of accessibility
150	3.7. Evolution of inclusion, diversity, or accessibility
12	3.8. Lack of evolution of inclusion, diversity, or accessibility
1.281	4. Research, education, creativity, and ICT
50	4.1. Promotion of research
286	4.2. Educational and innovative approach
114	4.3. Fostering creativity
230	4.4. Promotion of artistic quality
105	4.5. Promotion of ICT
87	4.6. High expectations
317	4.7. Motivation
14	4.8. Demotivation
78	4.9. Positive reinforcement
739	5. Projection and sustainability
69	5.1. Long-term vision and sustainability
190	5.2. Promotion and recommendation
480	5.3. Social transformation
1.121	6. Evaluation of the experience
236	6.1. Enjoyment and value of the musical experience
527	6.2. Positive experience assessment
35	6.3. Negative experience assessment
77	6.4. Assessment of project management and direction
115	6.5. Assessment of external projection
131	6.6. Expression of gratitude
492	7. Perspective or critical reflection
278	7.1. Challenges
214	7.2. Improvement proposals

**Table 2 healthcare-12-01355-t002:** Coding of the nodes of emotional impact in the questionnaire to the choir.

N	Nodes
1.811	Choir questionnaire coding
102	Positive emotional impact
23	Negative emotional impact

**Table 3 healthcare-12-01355-t003:** Coding of the nodes of emotional impact in the questionnaire to the environment.

N	Nodes
821	Environment questionnaire coding
26	Positive emotional impact
4	Negative emotional impact

**Table 4 healthcare-12-01355-t004:** Coding of the nodes of emotional impact in focus groups.

N	Nodes
717	Focus group coding
67	Positive emotional impact
3	Negative emotional impact

**Table 5 healthcare-12-01355-t005:** Sub-nodes/categories related to the emotional impact nodes.

Positive Emotional Impact		
Public Performances: gathers participants’ perceptions of positive emotions derived from performing in front of an audience.	Outside the Rehearsal: collects the positive emotional experiences experienced by the choir members in activities derived from the project, but outside the official programming of the project.	Activities: Gather feedback on the contribution of the choir’s scheduled activities to the emotional well-being of participants.
Personal Interactions: Groups references to the positive emotional impact of interpersonal relationships within the choir.	Contributions/Productions: Brings together perceptions of how members’ creative and musical contributions enrich their experience in the choir.	From the Audience: collects perceptions of how audience reactions and feedback positively influence participants’ experience.
From the Project: discusses how the overall aspects and philosophy of the project contribute to the emotional well-being of the members.		
**Negative Emotional Impact**		
Personal Interactions: groups participants’ perceptions of negative impacts in the context of interpersonal relationships.	Overwhelm/Stress/Demand: Encompasses perceptions of stress or pressure experienced by choir members.	Sadness at Leaving: Gathers allusions regarding the negative emotions associated with leaving the choir.
Non-inclusion: encompasses appraisals related to experiences of feeling excluded within the group.	Self-perception: collects perceptions of low self-esteem or negative self-perception of members.	

## Data Availability

Data are available upon request to the corresponding author.

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
