# Peer review of "Exploratory Study on the Impact on Emotional Health Derived from Participation in an Inclusive Choir"

_healthcare, 2024, doi:10.3390/healthcare12131355_

Round 1

Reviewer 1 Report

Comments and Suggestions for Authors

First of all I want to congrat you for the study that os very interesting and provides additional support to this kind off group interventions. It's a complete approach to subjective wellbeing.

I just have two suggestions for you:

- You could detail better the procedure related to ethical considerations, it's no clear on research design the care that you have to respect this considerations once that you do not have the support off a ethics commission or board;

- You could detail the questionnaire construction, it's not completely clear what was questionned and what qualitative questions means on this case. It's also important to know the domains of the questions and how was the structure of the questionnaire.

Thanks for your attention

Author Response

Thank you for your thoughtful feedback and suggestions. I appreciate your attention to detail and commitment to improving the clarity and rigor of the research.

Here are my responses to your comments:

Ethical Considerations

In the article, we detail our approach to ethical considerations between lines 196 and 209. We obtained informed consent from each participant for all instruments used in the study, and we anonymized their responses following international standards. These data were sent to the journal editor upon submission of the article. In the absence of an ethics committee at our university, we adhered strictly to the guidelines in the relevant literature to ensure rigorous ethical standards.

Text included:

In alignment with the journal's ethical policies, the research was conducted with the utmost responsibility and respect for the participants, strictly adhering to the highest ethical standards. All subjects provided informed consent prior to their participation. Throughout the study, participants' confidentiality and anonymity were preserved, following ethical principles of respect, beneficence, non-maleficence, return of results, and justice. The study meticulously followed protocols recommended by academic literature and the Declaration of Helsinki concerning research involving human subjects [25].

Moreover, it is important to highlight that the CIC, where the research was conducted, operates under the Music and Inclusion for Social Change Chair. This chair was established through a partnership between the University of Zaragoza and the Institute of Social Services of the Government of Aragon. This collaboration underscores a commitment to research integrity and ethics, particularly in projects promoting social inclusion through music and focusing on the wellbeing of vulnerable and disadvantaged individuals.

Questionnaire Construction

Regarding the construction of the questionnaire, the survey included demographic data of the respondents and covered dimensions such as perceived benefits, inclusion, social transformation, and challenges or areas for improvement. Specific questions included such as:

  • "In what ways do you think participation in the choir improves the lives of the participants, based on your own experience?"
  • "Does participating in the choir provide you with emotional growth?"

These questions were designed to capture qualitative data, offering a comprehensive understanding of the participants' experiences and perceptions. We have include this information in the paper.

Thank you again for your valuable feedback. Please let me know if there are any other areas where further clarification or detail might be beneficial.

Reviewer 2 Report

Comments and Suggestions for Authors

The article presented is extremely interesting and innovative. The text is clear, well developed and the methodology is relevant. On the other hand, the discussion is extremely brief and scarce, weakening the depth of the final work.

Therefore, this is an excellent work on emotions and their relationship with the field of music. However, the link between emotions and health is not clear. Emotions are basic neurological processes that are present in all people. Undoubtedly, the performance of certain activities can generate positive emotions. However, this does not imply that a positive emotion is inevitably healthy. Reading the paper, it seems that the authors start from this premise without providing any justification. In fact, it seems that in the introduction they suggest that the mere practice of music, in itself, is healthy (something that, as we know, is related to context, education, etc.). On the other hand, it does not seem clear that such an article has a place in a journal such as Healthcare as it does not conform to the journal's policies. In my opinion, they are not in the interdisciplinary area of aspects related to medicine and healthcare research. 

Author Response

We greatly appreciate your positive comments on our article. We are pleased to know that you found the text interesting, innovative, clear, and that you consider our methodology relevant.

Regarding your observation about the relationship between emotions and health, we would like to clarify that we do not start from the premise that any positive emotion is inherently healthy. Our focus is on investigating how participation in an activity like choral singing can influence participants' emotional health or emotional well-being. We do not suggest that the practice of music alone is healthy without considering context and other factors.

Our results include self-reported narratives from participants indicating specific improvements in their mental health due to the feelings and emotions generated by their participation in the choir. These narratives provide qualitative evidence of the emotional and well-being benefits experienced by participants, which we consider relevant to the field of emotional health.

Regarding the suitability of our article for a journal like Healthcare, we believe that our research aligns with the journal's objectives by exploring how cultural and artistic activities can positively impact people's emotional health. We consider that this approach provides a valuable perspective for medical and health research, integrating psychological and social aspects that are fundamental to overall well-being.

We thank you again for your comments and we have expand the discussion in the article to address the points raised in greater depth and to strengthen the connection between emotions, health, and the practice of music in specific contexts.

Reviewer 3 Report

Comments and Suggestions for Authors

The article titled "Exploratory Study on the Impact on Emotional Health Derived from Participation in an Inclusive Choir" aims to investigate the emotional health benefits and challenges associated with participation in an inclusive choir. The authors conducted a longitudinal exploratory study using a combination of participant observation, field notes, focus groups, and questionnaires to gain a comprehensive understanding of participants' emotional experiences.

Strengths

  1. Innovative Research Focus: The study addresses a unique and under-researched area, exploring the emotional impact of participation in an inclusive choir, which integrates individuals of diverse abilities and backgrounds. This focus on inclusivity and emotional health is commendable and timely.
  2. Comprehensive Methodology: The use of multiple qualitative methods, including participant observation, field notes, focus groups, and questionnaires, provides a rich and in-depth understanding of the participants' experiences. The longitudinal aspect of the study adds robustness to the findings.
  3. Positive Findings: The study highlights significant positive impacts on participants' emotional well-being, including increased self-esteem, a greater sense of belonging, and reduced symptoms of anxiety and depression. These findings underscore the therapeutic potential of inclusive choral activities.

Suggestions

  1. Inclusion of a Control Group: Future studies should consider including a control group to better establish the causal relationship between choir participation and emotional health improvements.
  2. Integration of Quantitative Methods: Combining qualitative and quantitative methods could provide a more comprehensive understanding of the impact of inclusive choir participation. Surveys with standardized measures of emotional health could complement the rich qualitative data.
  3. Focus on Negative Impacts: A more detailed exploration of the negative emotional impacts and their underlying causes could help develop strategies to mitigate these effects and enhance the overall positive experience of choir participants.

Overall, the article makes a valuable contribution to understanding the emotional health benefits of inclusive choir participation. With the suggested improvements, future research can build on these findings to further explore and validate the therapeutic potential of inclusive musical activities.

Author Response

Thank you very much for your thoughtful and encouraging feedback on our article. We are pleased to hear that you found the research focus innovative and the methodology comprehensive.

Inclusion of a Control Group: We agree that including a control group in future studies would enhance the ability to establish a causal relationship between choir participation and emotional health improvements. This is a valuable suggestion that we will consider for our subsequent research designs.

Integration of Quantitative Methods: Your recommendation to integrate quantitative methods alongside our qualitative approach is appreciated. Including standardized measures of emotional health in surveys could indeed complement our qualitative data and provide a more comprehensive understanding of the impact of inclusive choir participation. We will explore this mixed-methods approach in future studies.

Focus on Negative Impacts: We acknowledge the importance of a more detailed exploration of any negative emotional impacts experienced by participants. Understanding the underlying causes of these effects could help develop strategies to mitigate them and enhance the overall positive experience. We will ensure that future research addresses this aspect more thoroughly.

We are grateful for your constructive feedback and for highlighting the strengths of our study. We believe that with the suggested improvements, future research can build on our findings to further explore and validate the therapeutic potential of inclusive musical activities. We are committed to continuing this line of research and to refining our methodology to address the valuable suggestions provided.

Thank you once again for your insights and support.

Round 2

Reviewer 2 Report

Comments and Suggestions for Authors

The authors have made an effort to improve the work and link it to the editorial line of the journal. The quality and clarity of the work remains excellent. I have noticed that the tables and images do not indicate the source, please change this.

Author Response

Thank you for your feedback and positive comments. We have made the necessary adjustments to indicate the source of all tables and images as requested.